# An illustration of multi-class roc analysis for predicting internet addiction among university students

**Nishat Tasnim Thity[1], Atikur Rahman[2]\*, Adisha Dulmini[3], Mst. Nilufar Yasmin[4], Rumana Rois[5]**

**1** Department of Statistics and Data Science, Jahangirnagar University, Dhaka, Bangladesh,
**2** Department of Statistics and Data Science, Jahangirnagar University, Dhaka, Bangladesh,
**3** Department of Business and Law, University of Wollongong, New South Wales, Australia, **4** Department of Statistics and Data Science, Jahangirnagar University, Dhaka, Bangladesh, **5** Department of Statistics and Data Science, Jahangirnagar University, Dhaka, Bangladesh

\* arahman@juniv.edu

## Abstract

The internet is one of the essential tools today, and its impact is particularly felt among university students. Internet addiction (IA) has become a serious public health issue worldwide. This multi-class classification study aimed to identify the potential predictors of IA by four severity levels among university students in Bangladesh. We used cross-sectional survey data from 424 university students from different universities in Bangladesh. Data was collected using a self-reported questionnaire, along with an IA test to assess addiction levels. We identified the important features related to IA using the Boruta algorithm. Predictions were made using different machine learning (ML) (decision tree (DT), random forest (RF), support vector machines (SVMs), and logistic regression (LR)) models. Their performance was assessed using confusion matrix parameters, receiver operating characteristics (ROC) curves, and k-fold cross-validation techniques for multi-class classification problems. The prevalence of severe IA was 3.77% among the participating university students in Bangladesh from July 15 to July 22, 2024. University students' backgrounds, depression, anxiety, stress, participation in physical activity, misbehaving with family members, memory loss symptoms, and being COVID-19-positive were selected as significant features for predicting IA. Overall, the RF (accuracy = 0.531, sensitivity = 0.200, specificity = 0.986, precision = 1.00, k-fold accuracy = 0.4858, micro-average area under curve (AUC) = 0.7798) more accurately predicted IA compared to other ML techniques. The ML framework for multi-class classification study can reveal significant risk factors and predict this behavioral addiction more precisely. It can help policymakers, stakeholders, and families better understand the situation and prevent this severe crisis by improving policy-making strategies, promoting mental health, and establishing

**Data availability statement:** All relevant data are within the paper and its Supporting Information file. The datasets used in this study are provided as supplementary materials and can be accessed without restriction.

**Funding:** The author(s) received no specific funding for this work.

**Competing interests:** The authors have declared that no competing interests exist.

effective university counseling services. Therefore, raising awareness among the younger generation and their parents about the predictors of IA is important.

---

## 1. Introduction

The internet has become an integral part of daily life but its excessive and uncontrolled use, particularly among young people, has surfaced as a significant public health issue [1, 2, 3]. Internet Addiction (IA) is defined as the inability to control the urge for excessive internet use. This often results in reduced offline activities and significant anxiety or aggression when internet access is restricted. It can also lead to persistent social and family disruptions due to problematic internet use [4,5]. IA is considered a behavioral addiction that meets the six core components of addiction: salience, tolerance, mood modification, withdrawal symptoms, conflict, and relapse [6,7]. This global phenomenon is also evident in Bangladesh, where the increasing number of internet users raises concerns about its impact on university students.

According to a report by the Bangladesh Telecommunication Regulatory Commission (BTRC), the total number of internet users in Bangladesh was 141.28 million at the end of May 2024 [8]. This number seems reasonable now, but the gradually increasing number of users could pose a significant problem for university students nationwide. Numerous studies indicate that university students' time on the Internet for academic and non-academic purposes significantly influences their future careers [9]. Compared to non-addicted users, those with IA spend approximately an additional 0.64 hours per session and 1.08 extra days per week online. This significantly exceeds the average internet usage among university students, highlighting the substantial behavioral differences between addicted and non-addicted users [10,11]. There is also a strong association between IA and common psychological problems such as depression, anxiety, and stress [12]. In the USA, studies have reported that 5–12% of university students are addicted to excessive internet use [13]. After the outbreak of the COVID-19 pandemic, the rise in IA among university students has negatively impacted their physical, psychological, and social lives, severely hampering their well-being [14]. Identifying and addressing the common factors of IA contributes to achieving the Sustainable Development Goal (SDG) 3, which focuses on ensuring good health and well-being by 2030. IA has been linked to mental health challenges such as depression, anxiety, and stress, which directly impact well-being and academic performance. Addressing IA through targeted interventions, mental health support, and awareness campaigns aligns with SDG 3's objective to reduce premature mortality from non-communicable diseases and promote mental well-being. By recognizing IA as a public health concern and implementing effective prevention strategies, this study supports global efforts to improve mental health outcomes among university students [15].

A significant number of research have concentrated on IA among university students, revealing alarming levels of addiction that result in negative consequences for both personal and academic life. These studies highlight that IA can lead to delays in

completing assignments, lack of sleep, and poor academic performance, as well as impairments in social interactions and increased levels of anxiety, depression, and stress [1,3,10,16–21]. In Bangladesh, numerous studies have identified a strong relationship between IA and mental health issues. Research indicates that IA among Bangladeshi university students is associated with higher levels of psychological distress, including symptoms of depression, anxiety, and stress. Additionally, it has been observed that IA can lead to social withdrawal, decreased physical activity, and disturbances in sleep patterns [22–27]. In this study, we aim to identify the association between various socio-demographic characteristics and the prevalence of IA among university students in Bangladesh using the most effective Machine Learning (ML) model. ML techniques are particularly effective for capturing complex patterns in behavioral and psychological data, making them well-suited for predicting IA severity. Unlike traditional statistical methods, ML algorithms can model sophisticated relationships between multiple risk factors, allowing for a more detailed and comprehensive understanding of IA predictors. By applying ML, this study enhances predictive accuracy and provides valuable insights that can inform early intervention strategies and targeted mental health initiatives for university students. Therefore, we applied various well-known ML techniques, including decision tree (DT), random forest (RF), support vector machine (SVM), and logistic regression (LR), and evaluated their performances to determine the best-suited model for predicting the potential risk factors of IA among Bangladeshi university students.

## 2. Materials and methods

### 2.1 Data sources and study design

We conducted a cross-sectional online-based study among university students in Bangladesh from July 15 to July 22, 2024, using a snowball sampling approach. Participants were initially recruited through convenience sampling, after which they were encouraged to invite acquaintances to participate, forming a snowball sampling chain. Data were collected through a self-reported online survey, allowing participation from universities located in different urban and rural areas, enhancing the generalizability of the findings. The sample included public, private, technical, and medical university students across different regions of Bangladesh, ensuring a diverse representation. The study also covered students from various academic disciplines, including science, arts, commerce, medical, and engineering backgrounds. A total of 424 individuals (290 males and 134 females) from various universities participated, with responses collected via a well-structured Google form, ensuring no incomplete questionnaires. The response variable IA was categorized into four levels: Normal (0), Mild (1), Moderate (2), and Severe (3). Additionally, severe IA was recorded as a binary response (Yes = 1, No = 0). This methodology ensures a robust analysis of the prevalence and predictors of IA among Bangladeshi university students.

**Behavioral and demographic factors.** The input variables include various behavioral and demographic factors such as gender, age, academic year, educational background, university affiliation, marital status, involvement in physical activity and household chores, smoking habits, etc. The variable 'university affiliation' refers to the type of university attended by the participants (General, Private, Technical, or Medical). Meanwhile, 'involvement in household chores' indicates the participants engage in domestic tasks, categorized as Always, Sometimes, or Never. Body Mass Index (BMI), classified according to World Health Organization guidelines (WHO, 2010): underweight ($BMI < 18.5\ kg/m^2$), normal weight ($18.5\ kg/m^2 \leq BMI \leq 24.9\ kg/m^2$) and overweight/obese ($BMI \geq 25\ kg/m^2$) [28].

**IA Test (IAT).** To assess IA, we utilized the IAT scale developed by Young in 1998 [2]. The IAT is a widely recognized and validated tool for assessing internet addiction across various populations. Its applicability in Bangladesh is supported by prior research [26,27]. The questionnaire comprises 20 items, each scored on a 5-point Likert scale from "not applicable" (1) to "always" (5). The total IAT score ranges from 30 to 100. Participants were categorized into four groups based on their scores: normal internet usage (scores $\leq$ 30), mild IA (31–49), moderate IA (50–79), and severe IA (scores $\geq$ 80). Additionally, for a more intensive analysis of severe addiction, participants were further classified into severe internet users (scores $\geq$80) and non-severe internet users (scores < 80).

**Depression, Anxiety, and Stress Scale (DASS-21).** DASS-21 is a widely recognized self-report scale designed to assess depression, anxiety, and stress [29]. It consists of 21 items, with each subscale containing seven questions

rated on a four-point Likert scale (0 = never to 3 = almost always). In this study, scores were categorized into two groups: moderate to severe and mild to normal. The specific thresholds used were: depression (≥14), anxiety (≥10), and stress (≥19). These classifications help identify university students experiencing significant psychological distress. Prior research suggests that individuals with moderate to severe levels of depression, anxiety, and stress are more likely to develop IA, as excessive internet use can serve as a coping mechanism for underlying mental health challenges [29]. This association is particularly concerning among university students, who may use the internet to escape academic stress, social pressures, or emotional distress. Therefore, understanding the psychological risk factors linked to IA is essential for developing targeted mental health interventions.

**Ethics approval and consent to participate.** The study was approved by the Biosafety, Biosecurity, and Ethical Committee of Jahangirnagar University (Approval Number: Ref No: BBEC, JU/M 2024/ 07 (120)) and was conducted online, and data were collected with complete confidentiality and reliability. Participants did not receive any economic benefit, and anonymity was fully maintained. Verbal consent was obtained from all participants in an online format. Participants provided their consent by clicking a checkbox after being presented with detailed information about the study. The ethics committee approved this verbal consent procedure, ensuring it was appropriate for the online study format and that participants were adequately informed before consenting. The study retains full conformity with the international ethical guidelines for biomedical research on human participant research.

## 2.2 Statistical analyses

This study aimed to classify and predict potential predictors with IA, a multi-class situation, using different ML models (DT, RF, SVM, and LR). Our methodology involves collecting and pre-processing data and selecting features (the risk factors) using the Boruta algorithm. The evaluation process splits the entire data set into training and test data sets, applying ML models in the training data set and evaluating the performance of these models on the test data set. Therefore, the best-performed model predicts IA among university students based on the entire data set. Performance evaluations were conducted using three parameters from the confusion matrix: sensitivity, specificity, and accuracy, the area under the receiver operating characteristics (ROC) curve (AUC), and the K-fold cross-validation. All ML models were implemented using the scikit-learn module in Python's version 3.7.3. The Boruta algorithm, used for feature selection, was implemented using the Boruta package in the R programming language.

**2.2.1 Feature selection.** The chi-square ($\chi^2$) test is used to ensure the selection of statistically significant categorical features of IA. It evaluates the strength of the association between categorical predictor variables and the target variable, helping to identify features with meaningful relationships. Boruta algorithm is also employed to identify relevant risk factors for severe internet users among university students on the survey dataset. Unlike traditional feature selection methods, Boruta is an all-relevant feature selection technique that preserves both strong and potentially important but weaker predictors. As a wrapper method around the RF classification algorithm [30], it reduces the misleading impact of random fluctuations and correlations by introducing controlled randomness and aggregating results from multiple randomized samples. This process provides a more robust identification of fundamental attributes, ensuring that less relevant features are effectively removed [31]. Combining chi-square and Boruta methods strengthens the feature selection process. This integration ensures the model retains the most meaningful and relevant features, improving interpretability and predictive performance.

**2.2.2 ML models.** We implemented four different ML techniques: DT, RF, SVM, and LR. The DT is a widely used ML model that generates prediction algorithms for a target variable [32,33]. It classifies the population into branch-like segments shaped like an inverted tree with roots, leaf nodes, and branches [33]. In this tree-like structure, the internal nodes represent the features and attributes of the data, the branches represent the decision rules, and the leaf nodes represent the outcomes [34]. The DT algorithm can incorporate both nominal and numeric attributes [35]. RF are ensemble learning method used for classification and regression. They generate multiple DTs during training,

with predictions based on the mode (classification) or mean (regression) of the individual trees [36]. Each tree relies on random variables [37], aiming to minimize expected loss via a loss function [38]. SVM is used for both classification and regression problems, rooted in statistical learning theory [39]. They generate an optimal hyperplane that separates classes in the training data, maximizing the distance between the classes [40]. The solution is closely associated with support vectors, which define the hyperplane [41]. SVMs have shown strong performance, especially in classification tasks. LR is a statistical and ML model used for binary classification problems. LR determines the relationship between the categorical dependent variable and explanatory variables by maximizing the likelihood function [42]. LR does not assume normally distributed independent variables, making it suitable for a wide range of classification problems [43]. In this study, we used LR to analyze the association between various behavioral and demographic predictors and IA classification.

Using the GridSearchCV class from sklearn.model_selection in Python, hyperparameter tuning was used for the DT, RF, SVM, and LR models to determine the tuned parameter for each model. For instance, tuned hyperparameters for DTs include criteria = 'gini', max_depth = 10, min_samples_split = 5, max_leaf_nodes = 2, and max_features ='sqrt'; the RF model includes n_estimators = 1000 and max_features ='sqrt'; the SVMs include Tol = 0.01, max_iter = 40, C = 0.2, and Solver = 'saga'; and the LR model contains Tol = 0.01, max_iter = 40, Gamma = 'Scale', and Kernel = 'Sigmoid'.

**2.2.3 Predictive performances parameters.** We evaluated the predictive performance of ML models using key classification metrics derived from the confusion matrix, including accuracy, sensitivity (recall), specificity, and precision. Accuracy measures the proportion of correctly classified instances and is calculated as $accuracy = \frac{TP+TN}{TP+FP+TN+FN}$. Sensitivity (Recall) evaluates the model's ability to correctly identify positive cases, given by $sensitivity = \frac{TP}{TP+FN}$. Specificity measures how well the model correctly identifies negative cases, expressed as $specificity = \frac{TN}{TN+FP}$. Precision determines the proportion of correct positive predictions among all predicted positives and is defined as $precision = \frac{TP}{TP+FP}$. Here, TP (True Positives) refers to correctly predicted positive cases, TN (True Negatives) refers to correctly predicted negative cases, FP (False Positives) represents incorrect positive predictions, and FN (False Negatives) refers to incorrect negative predictions [44].

**2.2.4 Multi-class ROC.** Additionally, we assessed model performance using the Receiver Operating Characteristic (ROC) curve, which plots the True Positive Rate (Sensitivity) against the False Positive Rate (1 – Specificity) [45,46]. The Area Under the Curve (AUC) provides a summary measure of classifier performance, where a higher AUC value indicates better classification ability. For multi-class classification, separate ROC curves were generated for each class, and the individual class ROC curves were averaged using the macro-average method or combined with the micro-average method. Macro-average calculates individual class-wise AUC, treating all classes equally. Micro-average aggregates the contributions of all classes into a global confusion matrix and then computes the metric. This method accounts for class imbalance and gives more importance to the performance of the majority of classes [47].

**2.2.5 K-fold cross-validation.** K-fold cross-validation is a subsampling technique where the dataset is randomly split into k parts. In each iteration, k-1 parts of the dataset are used as the training set to build a model, and the remaining part is used as the test set to evaluate the model's accuracy. This process is repeated k times, with each part serving as the test set exactly once. The model with the smallest average cross-validation score is selected as the final model [48]. Each part is mutually exclusive, ensuring that the training and test sets in each iteration are disjointed [49]. As training folds will be closer to the entire dataset, a larger K reduces the bias towards overestimating the true expected error, but it also increases variance and running time. The $k = 10$ standard choices for medium-sized datasets balances performance and computation, while $k = 20$ and $k = 30$ are used for large datasets with critical model stability but increased computational cost [49]. Hence, we used $k = 10$, 20, and 30 for getting a comprehensive understanding of the model's performance across different conditions. Moreover, our study data is an imbalanced dataset, therefore, Stratified K-Fold Cross-Validation is more appropriate than Standard K-Fold Cross-Validation for this analysis. In stratified K-fold cross-validation, the dataset is split into k folds, but the splits are made in such a way that each fold maintains the same proportion of target classes as the original dataset. This ensures that each fold is representative of the overall class distribution. Stratified

 

k-fold cross-validation enhances model robustness and mitigates overfitting by rigorously evaluating performance across diverse data subsets, particularly crucial for imbalanced datasets [50,51].

## 3. Results

A total of 424 university students participated in an online survey-based study from different universities across Bangladesh. The frequency and percentage distributions of exposure variables and the prevalence of IA are presented in Table 1. A significant proportion of responses came from public university students (49.8%) and undergraduate university students (62.7%). Among the participants, 290 were male (68.3%) and 134 were female (31.6%). Additionally, 10.8% of the university students were smokers, 73.1% were currently unemployed, and 23.8% were overweight or obese. The study revealed concerning levels of mental health issues among university students: 66% reported depression, 60.1% suffered from anxiety, and 52.3% experienced stress. These rates are alarming considering the importance of mental health among university students. Furthermore, 53% of participants sometimes misbehave with family members, and 61.3% reported symptoms of memory loss. Almost all participants (99%) knew COVID-19, and 12% had contracted the disease. IA was slightly more severe among male university students (3.8%) than females (3.7%). University students with moderate to severe IA were more likely to be undergraduates, single, smokers, and not participate in physical or household activities. They also tend to misbehave with family members, suffer from memory loss symptoms, and experience depression, anxiety, and stress.

### 3.1 Features selection

Ten variables were identified as significantly associated with IA using the conventional chi-square test, as presented in Table 1. However, for some variables, the cell frequencies were less than 5, which can affect the reliability of the chi-square test. In these instances, we employed Fisher's exact test to ensure accurate variable selection [52,53]. These variables include University Category, Background, Smoking Status, Depression, Anxiety, Stress, Participation in Physical Activity, Misbehavior with family members, Memory Loss Symptoms, and COVID-19 Positive status. Fig 1 illustrates the main risk predictors of IA identified using the Boruta algorithm. The important features were extracted based on the mean decrease in accuracy. The key predictors identified were University Category, Background, Smoking Status, Depression, Anxiety, and Stress. The variable "Misbehavior with family members" was tentatively selected for predicting IA among university students. The variable that was identified using the chi-square test was also encompassed with the Boruta algorithm. Selected important possible predictors (using the Boruta algorithm and chi-square test) of IA were considered to assess the performances of various ML techniques using the parameters of a confusion matrix.

### 3.2 ML models evaluation

The performance of ML models is frequently assessed in binary classification problems. Such evaluation is rare in multi-class classification problems. This study is an example of the multi-class classification problem. According to Fig 2, among the participants, 3.77% were classified as severely addicted to the internet, 37.30% as moderately addicted, 34.67% as having mild IA, and 24.53% as having normal IA. Consequently, this task exemplifies an imbalanced multi-class assignment. To assess the overall performances of different ML techniques for predicting multi-levels of IA we used the selected predictors obtained from the Boruta algorithm and chi-square test, for instance, University Category, Background, Smoking Status, Depression, Anxiety, Stress, Participation in Physical Activity, Misbehavior with family members, Memory Loss Symptoms, and COVID-19 Positive status. Using the grid search optimization technique, hyperparameter tuning was performed for each ML model to enhance the model that yields the highest accuracy.

**Table 1. Frequency distribution and relationship with the prevalence of IA among university students.**

| Variables | IA | | | | $\chi^2$(*p*-value) | Fisher's Exact Test (*p*-value) |
|---|---|---|---|---|---|---|
| | Normal (%) | Mild (%) | Moderate (%) | Severe(%) | | |
| **Gender** | | | | | | |
| Male | 70 (24.1) | 98 (33.8) | 111 (38.3) | 11 (3.8) | 0.647 (0.886) | 0.650 (0.552) |
| Female | 34 (25.4) | 49 (36.6) | 46 (34.3) | 5 (3.7) | | |
| **Age** | | | | | | |
| 18-23 | 47 (24.0) | 66 (33.7) | 74 (37.8) | 9 (4.9) | 14.939 (0.245) | 14.355 (0.279) |
| 24-29 | 46 (23.6) | 70 (35.9) | 75 (38.5) | 4 (2.1) | | |
| 30-35 | 10 (38.5) | 8 (30.8) | 6 (23.1) | 2 (7.7) | | |
| 36-41 | 1 (25.0) | 2 (50.0) | 0 (0) | 1 (25.0) | | |
| 42-47 | 0 (0) | 1 (33.3) | 2 (66.7) | 0 (0) | | |
| **University Category** | | | | | | |
| General | 39 (18.5) | 87 (41.2) | 77 (36.5) | 8 (3.8) | 17.410 (0.043) | 18.004 (0.035) |
| Private | 30 (27.3) | 37 (33.6) | 38 (34.5) | 5 (4.5) | | |
| Technical | 32 (33.0) | 21 (21.6) | 41 (42.3) | 3 (3.1) | | |
| Medical | 3 (50) | 2 (33.3) | 1 (16.7) | 0 (0) | | |
| **Education Level** | | | | | | |
| Graduate | 37 (23.4) | 61 (38.6) | 56 (35.4) | 4 (2.5) | 2.453 (0.484) | 2.502 (0.475) |
| Undergraduate | 67 (25.2) | 86 (32.3) | 101 (38.0) | 12 (4.5) | | |
| **Background** | | | | | | |
| Science | 40 (18.9) | 90 (42.5) | 75 (35.4) | 7 (3.3) | 26.443 (0.009) | 25.805 (0.011) |
| Arts | 7 (24.1) | 7 (24.1) | 13 (44.8) | 2 (6.9) | | |
| Commerce | 6 (16.7) | 7 (19.4) | 21 (58.3) | 2 (5.6) | | |
| Medical | 3 (50.0) | 2 (33.3) | 1 (16.7) | 0 (0.0) | | |
| Engineering | 48 (34.0) | 41 (29.1) | 47 (33.3) | 5 (3.5) | | |
| **Marital status** | | | | | | |
| Married | 16 (27.6) | 22 (37.9) | 18 (31.0) | 2 (3.4) | 1.133 (0.769) | 1.152 (0.764) |
| Single | 88 (24.0) | 125 (34.2) | 139 (38.0) | 14 (3.8) | | |
| **No of Child of the Respondents** | | | | | | |
| 0 | 99 (24.6) | 138 (34.2) | 151 (37.5) | 15 (3.7) | 4.922 (0.554) | 4.374 (0.626) |
| 1 | 3 (20.0) | 7 (46.7) | 5 (33.3) | 0 (0.0) | | |
| 2 or More | 2 (33.3) | 2 (33.3) | 1 (16.7) | 1 (16.7) | | |
| **Smoking Status** | | | | | | |
| Yes | 5 (10.9) | 11 (23.9) | 24 (52.2) | 6 (13.0) | 20.593 (<0.001) | 17.411 (0.001) |
| No | 99 (26.2) | 136 (36.0) | 133 (35.2) | 10 (2.6) | | |
| **Profession** | | | | | | |
| Government Job | 9 (28.1) | 7 (21.9) | 16 (50.0) | 0 (0.0) | 11.106 (0.269) | 12.193 (0.203) |
| Private Sector | 17 (23.6) | 26 (36.1) | 25 (34.7) | 4 (5.6) | | |
| Business | 5 (50.0) | 3 (30.0) | 1 (10.0) | 1 (10.0) | | |
| University Students | 73 (23.5) | 111(35.8) | 115 (37.1) | 11 (3.5) | | |
| **Depression** | | | | | | |
| No | 65 (45.1) | 53 (36.8) | 25 (17.4) | 1 (0.7) | 66.308 (<0.001) | 68.445 (<0.001) |
| Yes | 39 (13.9) | 94 (33.6) | 132 (47.1) | 15 (5.4) | | |
| **Anxiety** | | | | | | |
| No | 78 (46.2) | 56 (33.1) | 32 (18.9) | 3 (1.8) | 81.586 (<0.001) | 83.669 (<0.001) |
| Yes | 26 (10.2) | 91 (35.7) | 125 (49.0) | 13 (5.1) | | |

*(Continued)*

| Variables | IA | | | | χ²(*p*-value) | Fisher's Exact Test (*p*-value) |
|---|---|---|---|---|---|---|
| | Normal (%) | Mild (%) | Moderate (%) | Severe(%) | | |
| **Stress** | | | | | | |
| No | 82 (40.6) | 81 (40.1) | 38 (18.8) | 1 (0.5) | 89.441 (<0.001) | 96.013 (<0.001) |
| Yes | 22 (9.9) | 66 (29.7) | 119 (53.6) | 15 (6.8) | | |
| **Way of Communication** | | | | | | |
| Direct call | 51 (24.9) | 72 (35.1) | 71 (34.6) | 11 (5.4) | 3.324 (0.344) | 3.379 (0.337) |
| Mobile application | 53 (24.2) | 75 (34.2) | 86 (39.3) | 5 (2.3) | | |
| **Participation in Physical Activity** | | | | | | |
| Always | 22 (34.9) | 19 (30.2) | 20 (31.7) | 2 (3.2) | 13.185 (0.040) | 13.248 (0.039) |
| Sometimes | 66 (22.0) | 116 (38.7) | 108 (36.0) | 10 (3.3) | | |
| Never | 16 (26.2) | 12 (19.7) | 29 (47.5) | 4 (6.6) | | |
| **Participation in household chores** | | | | | | |
| Always | 33 (27.0) | 43 (35.2) | 41 (33.6) | 5 (4.1) | 3.629 (0.727) | 3.776 (0.707) |
| Sometimes | 61 (22.8) | 96 (36.0) | 100 (37.5) | 10 (3.7) | | |
| Never | 10 (28.6) | 8 (22.9) | 16 (45.7) | 1 (2.9) | | |
| **Misbehave with family members** | | | | | | |
| Always | 0 (0.0) | 1 (16.7) | 4 (66.7) | 1 (16.7) | 36.620 (<0.001) | 37.112 (<0.001) |
| Sometimes | 33 (14.7) | 82 (36.4) | 99 (44.0) | 11 (4.9) | | |
| Never | 71 (36.8) | 64 (33.2) | 54 (28.0) | 4 (2.1) | | |
| **Memory loss symptoms** | | | | | | |
| Always | 3 (9.7) | 10 (32.3) | 15 (48.4) | 3 (9.7) | 15.458 (0.017) | 15.389 (0.017) |
| Sometimes | 58 (22.3) | 90 (34.6) | 105 (40.4) | 7 (2.7) | | |
| Never | 43 (32.3) | 47 (35.3) | 37 (27.8) | 6 (4.5) | | |
| **Participation in Religious activity** | | | | | | |
| Always | 44 (23.2) | 72 (37.9) | 66 (34.7) | 8 (4.2) | 4.793 (0.571) | 4.244 (0.644) |
| Sometimes | 55 (25.7) | 70 (32.7) | 83 (38.8) | 6 (2.8) | | |
| Never | 5 (25.0) | 5 (25.0) | 8 (40.0) | 2 (10.0) | | |
| **Waking up for bad dreams** | | | | | | |
| Always | 2 (11.8) | 4 (23.5) | 10 (58.8) | 1 (5.9) | 7.769 (0.256) | 7.566 (0.272) |
| Sometimes | 75 (23.1) | 117 (360) | 121 (37.2) | 12 (3.7) | | |
| Never | 27 (32.9) | 26 (31.7) | 26 (31.7) | 3 (3.7) | | |
| **Wash hands** | | | | | | |
| Always | 87 (25.8) | 122 (36.2) | 117 (34.7) | 11 (3.3) | 8.743 (0.189) | 6.978 (0.323) |
| Sometimes | 16 (19.5) | 24 (29.3) | 38 (46.3) | 4 (4.9) | | |
| Never | 1 (20.0) | 1 (20.0) | 2 (40.0) | 1 (20.0) | | |
| **Use Mask** | | | | | | |
| Always | 78 (24.0) | 111 (34.2) | 123 (37.8) | 13 (4.0) | 2.514 (0.867) | 2.167 (0.904) |
| Sometimes | 24 (26.7) | 33 (36.7) | 31 (34.4) | 2 (2.2) | | |
| Never | 2 (22.2) | 3 (33.3) | 3 (33.3) | 1 (11.1) | | |
| **Maintain Social Distance** | | | | | | |
| Always | 58 (29.6) | 57 (29.1) | 74 (37.8) | 7 (3.6) | 8.286 (0.218) | 8.399 (0.210) |
| Sometimes | 40 (19.3) | 82 (39.6) | 77 (37.2) | 8 (3.9) | | |
| Never | 6 (28.6) | 8 (38.1) | 6 (28.6) | 1 (4.8) | | |
| **COVID −19 Positive** | | | | | | |
| Yes | 14 (27.5) | 19 (37.3) | 13 (25.5) | 5 (9.8) | 7.985 (0.046) | 6.737 (0.081) |
| No | 90 (24.1) | 128 (34.3) | 144 (38.6) | 11 (2.9) | | |

*(Continued)*

**Table 1.** (Continued)

| Variables | IA | | | | $\chi^2$(p-value) | Fisher's Exact Test (p-value) |
|---|---|---|---|---|---|---|
| | Normal (%) | Mild (%) | Moderate (%) | Severe(%) | | |
| **Knowledge about COVID-19** | | | | | | |
| Yes | 103 (24.5) | 144 (34.3) | 157 (37.4) | 16 (3.8) | 3.542 (0.315) | 4.701 (0.195) |
| No | 1 (25.0) | 3 (75.0) | 0 (0.0) | 0 (0.0) | | |
| **Eat Nutritious Food** | | | | | | |
| Yes | 101 (24.8) | 143 (35.0) | 149 (36.5) | 15 (3.7) | 1.701 (0.637) | 1.651 (0.648) |
| No | 3 (18.8) | 4 (25.0) | 8 (50.0) | 1 (6.3) | | |
| **Take Vitamin Supplements** | | | | | | |
| Yes | 27 (23.5) | 42 (36.5) | 40 (34.8) | 6 (5.2) | 1.314 (0.726) | 1.259 (0.739) |
| No | 77 (24.9) | 105 (34.0) | 117 (37.9) | 10 (3.2) | | |
| **Google Classroom Users** | | | | | | |
| Yes | 81 (23.1) | 123 (35.1) | 132 (37.7) | 14 (4.0) | 2.226 (0.527) | 2.162 (0.540) |
| No | 23 (31.1) | 24 (32.4) | 25 (33.8) | 2 (2.7) | | |
| **BMI** | | | | | | |
| Overweight | 27 (22.9) | 43 (36.4) | 41 (34.7) | 7 (5.9) | 3.601 (0.731) | 3.475 (0.747) |
| Normal weight | 68 (25.2) | 91 (33.7) | 104 (38.5) | 7 (2.6) | | |
| Underweight | 9 (25.0) | 13 (36.1) | 12 (33.3) | 2 (5.6) | | |

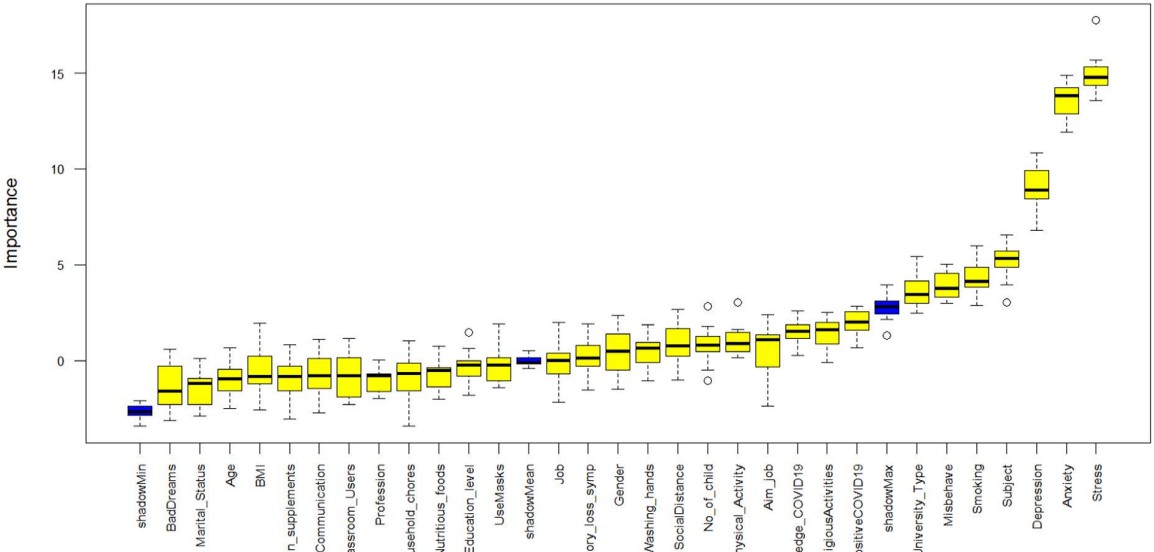

**Fig 1. Features selection using the Boruta algorithm.**

For this multi-class classification study, different ML models using 70% of observations as training data and 30% as test data, with a random seed of 99587, implemented using the scikit-learn module in Python. The confusion matrices obtained are explored in Fig 3, and their calculated different performance parameters are presented in Table 2. Column-wise all the cells of the confusion matrix for SVM and LR models were zero in Fig 3, hence, sensitivity and precision cannot be calculated for those models in Table 2.

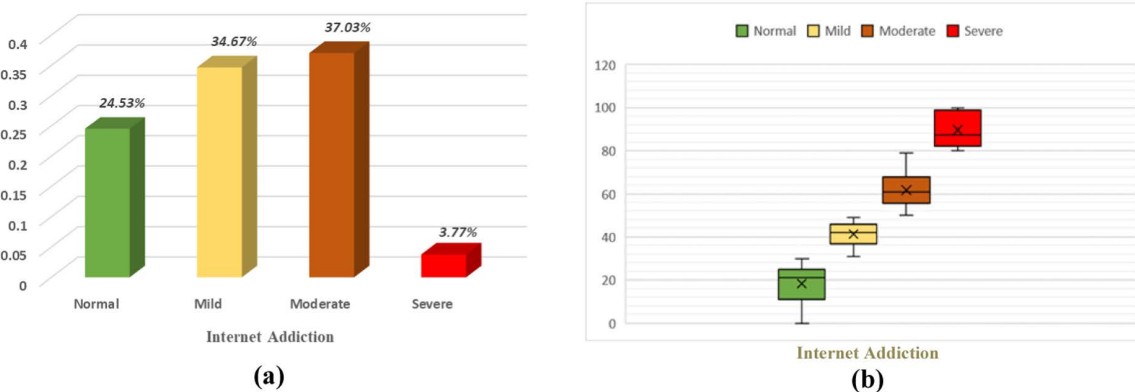

**Fig 2. Prevalence of IA by severity.** in (a) and the distribution of IA scores by severity in **(b)**, where the box indicates standard deviation, and the whisker indicates min – max range.

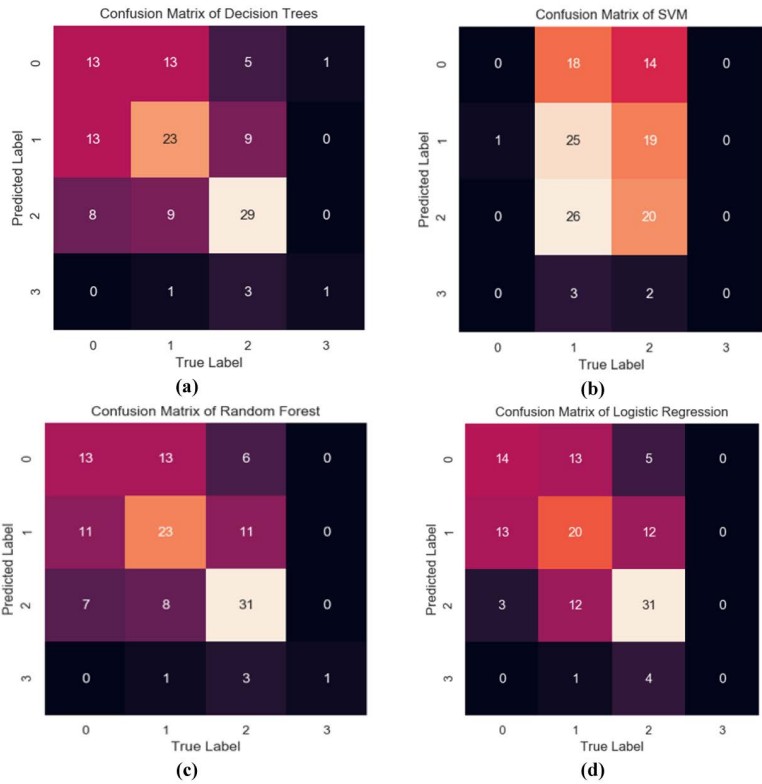

**Fig 3. The output of confusion matrices based on ML models.** DT in **(a)**, RF in **(b)**, SVM in **(c)**, and LR in **(d)**.

Table 2 reveals that the RF model emerged as the most efficient for predicting IA, providing 53.13% accurate predictions (accuracy = 0.5313), 20% sensitivity (positive cases correctly predicted), 98.6% specificity (negative cases correctly predicted), and 100% precision (correct positive predictions). Based on the results, the RF technique was found to be superior compared to other ML models for predicting the IA. The ROCs were calculated for this multi-class classification problem using the scikit-learn module for four ML techniques and are presented in Fig 4.

**Table 2. Accuracy, sensitivity, specificity, and precision of different ML models to predict multi-classes of IA among university students in Bangladesh.**

| Models | Accuracy | Sensitivity | Specificity | Precision |
|---|---|---|---|---|
| DT | 0.516 | 0.200 | 0.991 | 0.500 |
| RF | **0.531** | **0.200** | **0.986** | **1.00** |
| SVM | 0.352 | – | 0.960 | – |
| LR | 0.508 | – | 0.960 | – |

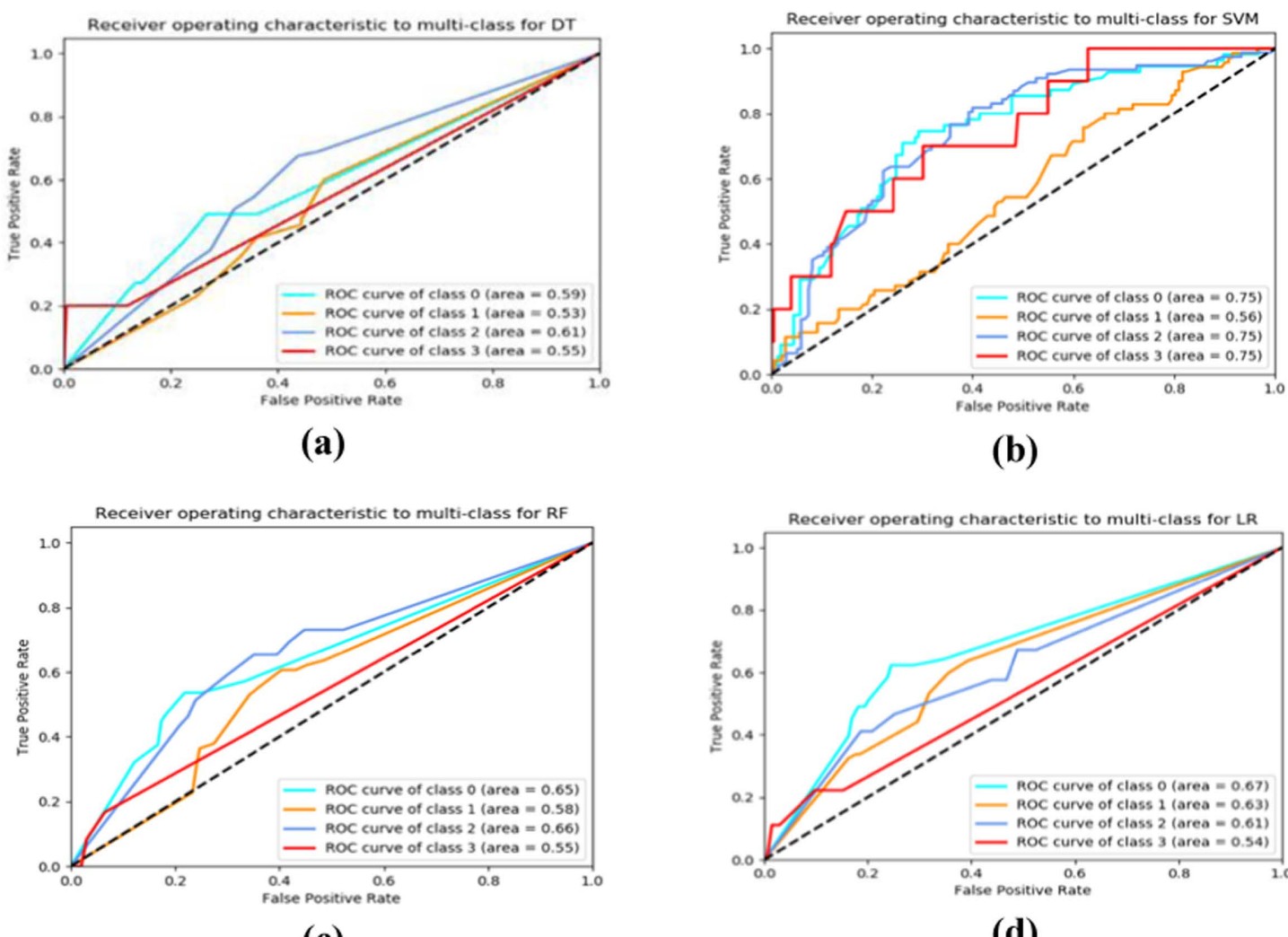

**Fig 4. The ROC curves to predict multi-classes of IA among university students in Bangladesh using ML models.** DT in **(a)**, RF in **(b)**, SVM in **(c)**, and LR in **(d)**.

The summary results of k-fold cross-validation for $k = 10$, 20, and 30 are shown in Table 3 using the ten selected significance factors based on the Boruta algorithm and the chi-square test. For implementing different ML models, we converted the categorical data to the count data using General=1, Medical=2, Private=3, Technical=4 for the variable University type; Background: Arts=1, Commerce=2, Medical=3, Engineering=4, and Science=5; and used Never=1, Sometimes=2, Always=3, Yes=1, No=0.

**Table 3. Result of K-fold cross-validation of ML models.**

| Models | 10-Fold | | 20-Fold | | 30-Fold | |
|---|---|---|---|---|---|---|
| | MAcc | SE | MAcc | SE | MAcc | SE |
| DT | 0.4575 | 0.0500 | 0.4855 | 0.0877 | 0.4732 | 0.1414 |
| RF | 0.4858 | **0.0446** | 0.4951 | **0.0777** | 0.4925 | **0.1319** |
| SVM | 0.4972 | 0.0686 | 0.5180 | 0.1074 | 0.5152 | 0.1396 |
| LR | **0.5135** | 0.0806 | **0.5182** | 0.1078 | **0.5224** | 0.1346 |

**MAcc**=Mean of Accuracy scores, **SE**=Standard Error of Accuracy scores

Fig 4 provides an illustration of multi-class ROC analysis to predict IA by four levels of IA. To predict the normal level (Class = 0) of IA among university students in Bangladesh, the estimated AUC scores were 0.59 for DT, 0.65 for RF, 0.75 for SVM, and 0.67 for LR, with SVM performing best. For predicting the mild level (Class = 1) of IA, the AUC scores were 0.53 for DT, 0.58 for RF, 0.56 for SVM, and 0.63 for LR, with LR performing best. For moderate level (Class = 2) IA, the AUC scores were 0.61 for DT, 0.66 for RF, 0.75 for SVM, and 0.61 for LR, with SVM again performing best. Finally, for severe level (Class = 3) IA, the AUC scores were 0.555 for DT, 0.555 for RF, 0.75 for SVM, and 0.54 for LR, with SVM showing the highest AUC. However, our findings show that the Micro-Average AUC for DT, RF, SVM, and LR is 0.7058, 0.7798, 0.6587, and 0.7749, respectively. The micro-average AUC accounts for class imbalance and places greater emphasis on performance in the majority classes. Therefore, based on the multi-class ROC analysis, RF model performed better with the highest micro-average AUC 0.7798. Table 3 shows that the LR model provided the highest accuracy score, but the RF model had the minimum standard error, indicating that RF performed better in 10-fold, 20-fold, and 30-fold cross-validation. Based on the accuracy measure, the ROC, and the k-fold cross-validation approaches, the RF model was found to be better compared to other ML techniques for predicting the prevalence of IA among Bangladeshi university students within the last 12 months.

### 3.3 RF model to predict IA

Fig 5 illustrates the feature contributions among the ten variables selected using chi-square tests, which included the seven variables identified by the Boruta algorithm. The contributions were highest for educational background (14.1%), memory loss symptoms (13.4%), physical activity (12.2%), university category (11.9%), stress (9.7%), misbehavior with family members (9.6%), and anxiety (9.1%). In contrast, the contributions were lower for depression (7.1%), COVID-19 positivity (6.8%), and smoking (6.1%).

Figs 6–8 visualize the top one to the top third of trees from the RF model. Each node contains five components: feature's question, Gini impurity, samples, value, and class, with the terminal leaf nodes containing four components: Gini impurity, samples, value, and class. The "Gini" part describes the Gini impurity of the node, showing the decrease in average weighted Gini impurity as the path moves down the tree. "Samples" indicates the number of observations in the node, "value" represents the number of samples in each class, and "class" indicates the majority classification for points in the node (the prediction for all samples in the leaf node). Each feature's question splits the node with a True (left) or False (right) answer. A data point moves down the tree, reaching a leaf node (the final decision) based on these answers. Blue-colored leaves indicate a prediction of IA, while orange-colored leaves indicate non-IA, as shown in Figs 6–8. Fig 6 shows the top tree from the RF model, Fig 7 is the second top tree, and Fig 8 is the third top tree. To predict any respondent's data, follow the path down the trees using the answers to the feature's questions until reaching a leaf node, where the class is the prediction.

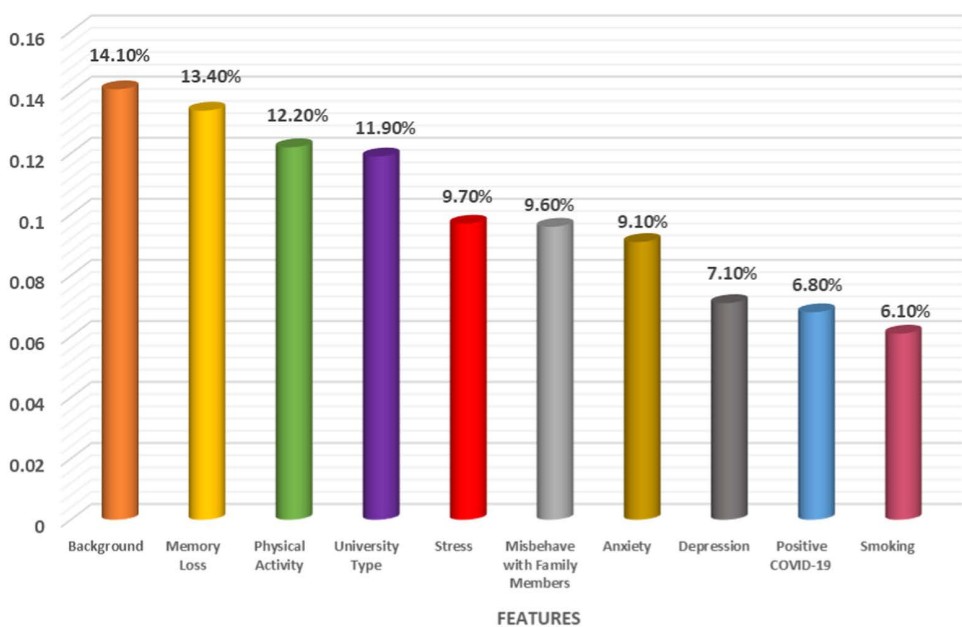

**Fig 5. Feature contributions for the selected features based on the RF model.**

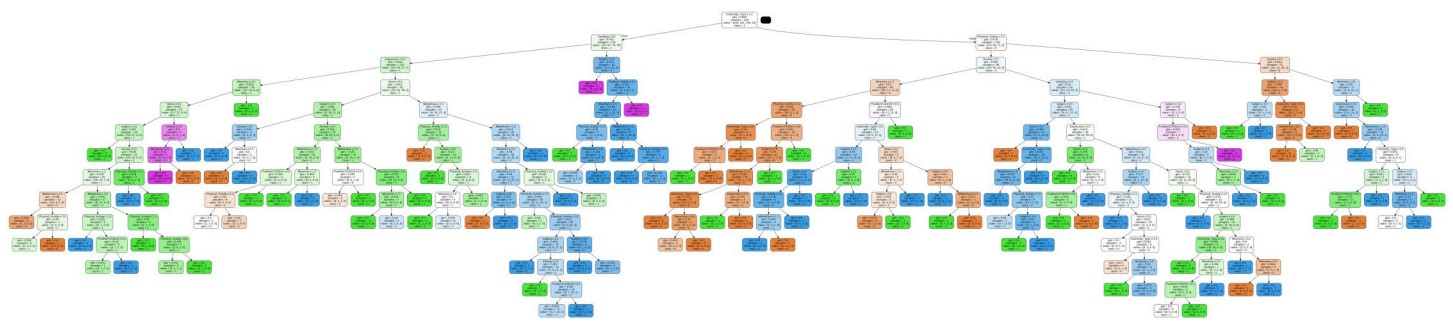

**Fig 6. Top one tree from the fitted RF model to predict multi-levels of IA.**

## 4. Discussion

University students play a crucial role in national development, as they contribute to academic advancements, technological innovations, and future workforce capabilities. However, IA can significantly delay their academic progress, cognitive development, and innovation potential, ultimately affecting national growth. IA has emerged as a major public health concern, particularly among university students, leading to various negative consequences [54,55]. Excessive internet use can cause deterioration in academic performance, reduced engagement in productive activities, and a decline in critical thinking and problem-solving skills, which are essential for workforce readiness and economic progress. Therefore, understanding the predictors of IA is essential for designing intervention strategies that promote balanced digital engagement while fostering intellectual growth.

This study identified several significant predictors of IA severity among university students using different ML techniques. Key predictors included university category, academic background, smoking status, depression, anxiety, stress,

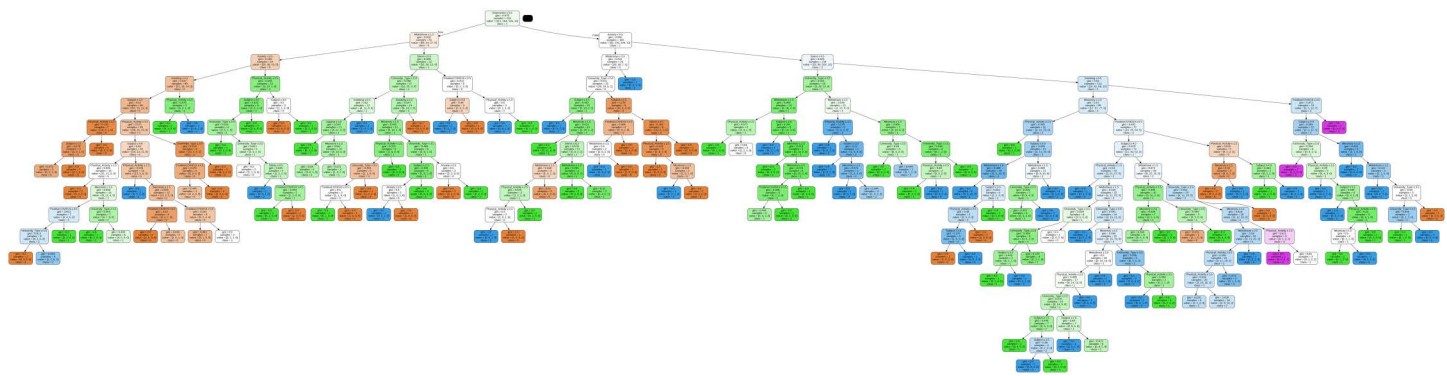

**Fig 7. Top 2ⁿᵈ tree from the fitted RF model to predict multi-levels of IA.**

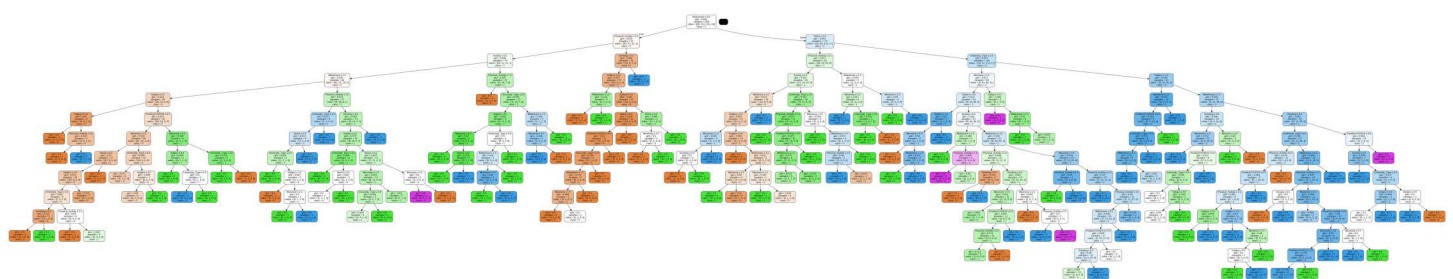

**Fig 8. Top 3ʳᵈ tree from the fitted RF model to predict multi-levels of IA.**

memory loss, and misbehavior with family members [10–12,56]. Each of these predictors has important implications for understanding IA. Psychological factors such as depression, anxiety, and stress often drive students to excessive internet use as a coping mechanism [14–16,57]. Conversely, IA can exacerbate mental health issues, creating a bidirectional relationship between psychological distress and problematic internet behavior. Cognitive and behavioral indicators, such as memory loss and misbehavior with family members, highlight the cognitive and emotional instability associated with IA, indicating a decline in cognitive functioning and social interactions, which further perpetuate internet dependency. Lifestyle factors like smoking status and physical activity also contribute to IA, as students who engage in unhealthy behaviors may be more likely to develop problematic internet habits. Furthermore, university affiliation plays a role, with students from specific categories (e.g., private universities) being more vulnerable to IA due to flexible schedules, academic pressure, and greater access to digital platforms. Certain student categories exhibit greater vulnerability to IA, reflecting both causes and consequences. For example, students with high academic stress may turn to the internet as an escape, while excessive internet use leads to further academic decline and stress. This cycle underscores the need for holistic interventions that address both the root causes and the outcomes of IA. Students from vulnerable backgrounds require targeted support to break this cycle and develop healthier digital habits. The use of the Boruta algorithm proved particularly effective for identifying key predictors in a multi-class setting. Unlike traditional feature selection methods, Boruta retains all relevant features, including weaker but potentially significant predictors. This approach was essential for our study, as IA severity exists on a spectrum rather than as a binary condition. By integrating Boruta with the chi-square test, we ensured robust feature selection that combined predictive relevance with statistical significance, enhancing the interpretability and performance of the models.

Our findings align with prior research demonstrating the effectiveness of RF and SVM in IA prediction. Previous studies have shown that RF performs well in handling high-dimensional datasets, while SVM exhibits strong classification accuracy for distinguishing IA severity levels [30,34]. However, the predictive accuracy of our models, which reached a maximum of 53.1%, reflects the challenges in modeling complex behavioral traits. This may be attributed to the complexity of IA, imbalanced data, and the presence of overlapping behaviors among different IA severity levels. Future studies should address these issues by employing advanced techniques and exploring more sophisticated models.

This study has several limitations. The use of a convenience sample may have led to selection bias, making the results potentially unrepresentative of the broader Bangladeshi university student population. Additionally, data collection through an online survey may have skewed the sample toward individuals more likely to exhibit IA, thereby limiting the generalizability of the findings. The recruitment method, where participants invited acquaintances, may have introduced homophily bias, as individuals with similar behaviors were more likely to be included. Furthermore, due to computational complexity and interpretability concerns, we utilized DT, RF, SVM, and LR models, while the relevance of other ML models such as Gradient Boosting Machines, K-Nearest Neighbors, and Neural Networks were not included. Future research should address these limitations by adopting a more diverse and representative sampling approach to improve the generalizability of findings. Additionally, longitudinal studies can help track IA trends over time and assess causal relationships. Incorporating more advanced ML techniques, such as deep learning and ensemble models, may enhance predictive accuracy. Furthermore, exploring intervention strategies using ML-based models could provide actionable insights for mitigating IA risks among university students.

## 5. Conclusion

The study results highlight a concerning level of internet addiction among university students in Bangladesh, which may pose risks to mental health, academic performance, and future professional careers. However, due to the use of a convenience snowball sample, these findings cannot be generalized to all Bangladeshi university students, and further research using more representative sampling methods is needed to assess the true prevalence and trends of internet addiction in this population. Recognizing the severity of this public health issue is crucial for identifying its underlying causes and implementing targeted interventions. Based on study findings universities can implement targeted awareness campaigns to educate university students about the risks of IA, particularly focusing on vulnerable groups. Incorporating mental health screenings and counseling services in educational settings can help address underlying psychological issues like depression and anxiety that contribute to IA. Additionally, promoting healthy digital habits and balanced internet use through workshops and structured programs can mitigate the negative impact of excessive internet use on university students' academic and personal development.

## Supporting information

**S1 Data. Data.**
(CSV)

## Acknowledgments

We appreciate all the participants as well as the anonymous reviewers.

## Author contributions

**Conceptualization:** Rumana Rois.
**Data curation:** Nishat Tasnim Thity, Atikur Rahman, Adisha Dulmini.
**Formal analysis:** Nishat Tasnim Thity, Adisha Dulmini.

**Methodology:** Atikur Rahman.

**Software:** Mst. Nilufar Yasmin.

**Supervision:** Rumana Rois.

**Visualization:** Adisha Dulmini, Mst. Nilufar Yasmin.

**Writing – original draft:** Nishat Tasnim Thity.

**Writing – review & editing:** Atikur Rahman, Mst. Nilufar Yasmin, Rumana Rois.

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
