## [Decision Letter · Decision Letter 0]

Dear Dr. Rahman,

Thank you for submitting your manuscript to PLOS ONE. After careful consideration, we feel that it has merit but does not fully meet PLOS ONE’s publication criteria as it currently stands. Therefore, we invite you to submit a revised version of the manuscript that addresses the points raised during the review process.

We look forward to receiving your revised manuscript.

Kind regards,

Enamul Kabir

Academic Editor

PLOS ONE

Journal Requirements:

3. In the online submission form, you indicated that the datasets that support the findings of this study are available on request. 

4. Please remove all personal information, ensure that the data shared are in accordance with participant consent, and re-upload a fully anonymized data set. 

Reviewers' comments:

Reviewer's Responses to Questions

**Comments to the Author**

1. Is the manuscript technically sound, and do the data support the conclusions?

Reviewer #1: Partly

Reviewer #2: Yes

2. Has the statistical analysis been performed appropriately and rigorously?

Reviewer #1: Yes

Reviewer #2: Yes

3. Have the authors made all data underlying the findings in their manuscript fully available?

Reviewer #1: Yes

Reviewer #2: Yes

4. Is the manuscript presented in an intelligible fashion and written in standard English?

Reviewer #1: Yes

Reviewer #2: Yes

Reviewer #1: Authors have done very well in the preparation of this manuscript and the execution of their study and I would like to congratulate them on their work. I also admire that the manuscript made an effort to explain the machine learning methods used to the readers and not simply assume they are familiar with them. I also found it interesting how two different feature selection methods were used.

Although, I have noticed some machine learning models that are applicable to multi-class outcome variables that were not fitted in this study. Namely, gradient boosting machines, K-nearest neighbour and neural networks. Is there a reason they were omitted? No reason was provided in the manuscript. I would advise that the authors either fit these models and incorporate the results of fitting these models into the paper, or provide concrete reasons why they cannot be. For instance, if they were omitted due to space or word constraints, or if they were omitted for technical reasons, this should be mentioned.

Furthermore, the authors do not mention the limitations of this study, when there are a few.

For instance, the fact that a convenience sample was used suggests that the results could be based on unrepresentative data. Many people are unwilling or unable to participate in surveys, after all. The fact the data was obtained from an online survey may have also skewed the sample towards containing participants more likely to have an internet addiction in the first place. Likewise, the fact that participants recruited their acquaintances to participate may have skewed the results too, as it is conceivable that people with signs of internet addiction are more likely to have friends that share this affliction.

The fact the models had, at best, a 53.1% prediction accuracy could also be discussed as a limitation. As could the limited number of models used, unless this is addressed by the authors in the next revision.

In section 2.1, there are also a couple of oversights. The categorization of weight status based on BMI scheme mentioned does not provide a reference, which it should. And I can find at least one source (https://www.who.int/data/gho/data/themes/topics/topic-details/GHO/body-mass-index?introPage=intro_3.html) that suggests that the World Health Organization uses BMI<18.5kg/m^2 as the cut-off for underweight, not BMI<20 kg/m^2. It also does not specify how participants' weight statuses were classified when BMI equalled the cut-off values of 20 kg/m^2 (below which was designed as underweight and above which was designed as normal weight) and 25 kg/m^2. Ideally, this manuscript should specify how people with a BMI of 20 kg/m^2 and 25 kg/m^2 were categorized, not just those with a BMI of above or below these values.

One question that arose while reading the study was why the multi-categorical outcome variable of internet addiction severity was used instead of the quantitative internet addiction score. Was this choice made purely for novelty, or were there other considerations? It is conceivable that since the internet addiction score variable can take on a broader range of different values, it may contain more information that the models can learn from than the internet addiction severity categorical variable, and hence it may have greater predictive power.

It is also important to note that the data does not support the first sentence of the Conclusion section. Namely,

"The study results highlight a concerning rise in IA among university students in Bangladesh, posing significant risks to mental health, academic performance, and future professional careers."

The use of a convenience snowball sample means that the data cannot reliably determine the prevalence of internet addiction among Bangladeshi university students or whether it is increasing.

The manuscript also does not discuss potential opportunities for further research arising from this work, which should ideally be included.

Lastly, it would be helpful to include a table comparing the demographic breakdown of the sample to that of all Bangladeshi university students at the time of the study (or at least the universities sampled). This would allow readers to assess how representative the sample is. This is, of course, assuming that data on the demographic breakdown of all Bangladeshi university students is available (such as from Bangladesh's statistics bureau, or from the country's department of education).

Reviewer #2: Dear Editor in the Plose ONE

Thank you for your invitation to review manuscript

Introduction

comment 1: The definition of Internet Addiction (IA) is comprehensive, but the sentence "This leads to minimizing offline time..." could be restructured for better readability. Consider breaking it into two sentences for clarity.

Example: "IA is defined as the inability to control the urge for excessive internet use. This often results in reduced offline activities and significant anxiety or aggression when internet access is restricted."

comment 2: The transition between the global context and Bangladesh-specific data could be smoother. Consider introducing the Bangladesh context with a linking sentence like, "This global phenomenon is also evident in Bangladesh, where..."

comment 3:The impact of IA on academic life is repeated in multiple places. You might consolidate these points for better focus and avoid redundancy.

comment 4: The comparison of internet usage time ("extra 0.64 hours per session...") could be more impactful with context or comparison to average usage.

comment 5: The connection between IA's effects and the Sustainable Development Goal (SDG) 3 is significant but could be made more prominent. Expand slightly on how addressing IA aligns with SDG 3 goals.

comment 6: The introduction of ML techniques is clear, but the connection between their use and the problem at hand could be stronger. Add a brief explanation of why ML techniques are uniquely suited for this analysis.

comment 7: Use consistent terminology, such as "university students" instead of alternating between "students" and "university students."

comment 8: Ensure all acronyms like ML, IA, DT, RF, SVM, and LR are clearly defined when first mentioned.

Data Sources and Study Design

Clarity: The use of "snowball sampling" and "convenient sampling" together might confuse readers. Clarify whether both techniques were employed or if these terms are used interchangeably.

Inclusivity of Data: Provide more details on the diversity of universities (e.g., public vs. private, regional distribution) to give context to the generalizability of findings.

IA Categorization: The binary classification for severe IA (Yes/No) is clear, but ensure consistency when explaining the four-level IA categorization.

Detail: Some variables, like "university affiliation" and "involvement in household chores," could benefit from further clarification.

While WHO standards are used, consider mentioning the cultural or regional relevance of BMI categories in Bangladesh for context.

Clarity: The explanation of score ranges is clear, but consider mentioning why Young's IAT was chosen and its cultural validity in Bangladesh.

Scoring: When explaining severe IA classification, avoid repeating information already stated. Instead, cross-reference earlier sections.

The thresholds for depression, anxiety, and stress are well-described, but their relevance to IA could be more explicitly linked.

While the mention of psychometric properties is important, providing specific reliability or validity coefficients for the Bangladeshi sample (if available) would strengthen this section.

The explanation of the Boruta algorithm is clear, but a brief rationale for its choice over other feature selection methods would be beneficial.

Clearly explain how the chi-square test complements the Boruta algorithm in the analysis pipeline.

The description of the four ML models is comprehensive but somewhat inconsistent in detail. For instance, logistic regression has less technical explanation compared to others.

The use of "1000 decision trees" in RF is good detail. However, mention if hyperparameter tuning (e.g., grid search or random search) was applied to optimize the models.

The formulas are essential but could be simplified visually with proper formatting. Consider presenting them as a table or inline equations for better readability.

The explanation is clear, but for multi-class classification, mention how individual class ROC curves were combined or averaged.

K-Fold Cross-Validation

Detail: Specify the value of "k" used in the cross-validation process (e.g., 5, 10) and explain why this value was chosen.

Clarity: Highlight how cross-validation ensures model robustness and avoids overfitting, especially when using imbalanced datasets.

Discussion

The opening effectively highlights the importance of university students to national development and the adverse effects of IA. However, the connection between IA and its long-term impact on growth could be elaborated.

Suggestion: Briefly explain how IA hinders academic and innovative capabilities.

The identification of significant predictors of IA is well-presented. However, it would help to briefly discuss the implications of each predictor.

Highlight why students from specific categories may be more vulnerable.

Mention how these factors could be both causes and effects of IA.

Briefly mention why this feature selection method was particularly useful for identifying predictors in a multi-class setting.

The performance of ML models is clearly described, but the discussion would benefit from:

Comparison with Literature: Mention if similar models (e.g., RF, SVM) have been used successfully in IA studies elsewhere.

Limitations: Acknowledge the relatively low accuracy (53.1%) of the RF model and discuss potential reasons (e.g., imbalanced data, complexity of IA).

Link findings to potential interventions or policies. For example:

Policy Implications: "Targeted awareness campaigns in universities, particularly for students from vulnerable backgrounds, could mitigate IA risks."

"Incorporating mental health screenings and counseling in educational settings may address underlying factors like depression and anxiety that contribute to IA."

The conclusion effectively emphasizes the rise in IA and its consequences. To strengthen it

While the ML framework is highlighted, consider elaborating on how ML can directly inform interventions

**Do you want your identity to be public for this peer review?** For information about this choice, including consent withdrawal, please see our Privacy Policy

Reviewer #1: **Yes: ** Brenton Horne

Reviewer #2: No

---

## [Author Response · Author response to Decision Letter 1]

21 Mar 2025

21 March 2025

An illustration of multi-class roc analysis for predicting internet addiction among university students

Dear Editor,

Thank you very much for sending us the editor and reviewer’s feedback and providing an opportunity to revise the manuscript. We thank the editor and the reviewers for their useful comments and suggestions. We have revised the paper as per the editor and reviewer’s comments and a response to reviewers comments are given below:

Best regards,

Authors.

Editor comments

• Please ensure that your manuscript meets PLOS ONE's style requirements, including those for file naming. The PLOS ONE style templates can be found at https://journals.plos.org/plosone/s/file?id=wjVg/PLOSOne_formatting_sample_main_body.pdf and https://journals.plos.org/plosone/s/file?id=ba62/PLOSOne_formatting_sample_title_authors_affiliations.pdf

Authors’ response

Thank you very much for the comment. We have revised the formatting according to the PLOS ONE style templates.

• Your ethics statement should only appear in the Methods section of your manuscript. If your ethics statement is written in any section besides the Methods, please move it to the Methods section and delete it from any other section. Please ensure that your ethics statement is included in your manuscript, as the ethics statement entered into the online submission form will not be published alongside your manuscript.

Author's response

We have moved the ethics statement to the Methods section (page # 7) and deleted it from the other section.

• In the online submission form, you indicated that the datasets that support the findings of this study are available on request. All PLOS journals now require all data underlying the findings described in their manuscript to be freely available to other researchers, either 1. In a public repository, 2. Within the manuscript itself, or 3. Uploaded as supplementary information.

Authors’ response

We have uploaded the data file Within the manuscript itself as supporting information.

• Please remove all personal information, ensure that the data shared are in accordance with participant consent, and re-upload a fully anonymized data set.

Authors’ response

All the personal information is removed, and the final dataset is attached as a supporting document.

Reviewer #1 comments

• Authors have done very well in the preparation of this manuscript and the execution of their study and I would like to congratulate them on their work. I also admire that the manuscript made an effort to explain the machine learning methods used to the readers and not simply assume they are familiar with them. I also found it interesting how two different feature selection methods were used.

Authors’ response

Thank you very much for the comment. We have added the clarification of using both chi-square and Boruta algorithms for feature selections on page # 9 section 2.2.1 Feature selection.

• Although, I have noticed some machine learning models that are applicable to multi-class outcome variables that were not fitted in this study. Namely, gradient boosting machines, K-nearest neighbour and neural networks. Is there a reason they were omitted? No reason was provided in the manuscript. I would advise that the authors either fit these models and incorporate the results of fitting these models into the paper, or provide concrete reasons why they cannot be. For instance, if they were omitted due to space or word constraints, or if they were omitted for technical reasons, this should be mentioned.

Authors’ response

Thank you for this valuable comment. We have added clarification as a limitation of the study in the Discussion section (page # 25). The Clarification is now: Gradient Boosting Machines, K-Nearest Neighbors, and Neural Networks were not included primarily due to computational complexity and interpretability concerns. However, we acknowledge their relevance and will consider them for future work.

• Furthermore, the authors do not mention the limitations of this study, when there are a few. For instance, the fact that a convenience sample was used suggests that the results could be based on unrepresentative data. Many people are unwilling or unable to participate in surveys, after all. The fact the data was obtained from an online survey may have also skewed the sample towards containing participants more likely to have an internet addiction in the first place. Likewise, the fact that participants recruited their acquaintances to participate may have skewed the results too, as it is conceivable that people with signs of internet addiction are more likely to have friends that share this affliction. The fact the models had, at best, a 53.1% prediction accuracy could also be discussed as a limitation. As could the limited number of models used, unless this is addressed by the authors in the next revision.

Authors’ response

Thank you very much for the comment. We have added the limitations of the study at the end of the discussion section page # 25.

• In section 2.1, there are also a couple of oversights. The categorization of weight status based on BMI scheme mentioned does not provide a reference, which it should. And I can find at least one source (https://www.who.int/data/gho/data/themes/topics/topic-details/GHO/body-mass-index?introPage=intro_3.html) that suggests that the World Health Organization uses BMI<18.5kg/m^2 as the cut-off for underweight, not BMI<20 kg/m^2. It also does not specify how participants' weight statuses were classified when BMI equalled the cut-off values of 20 kg/m^2 (below which was designed as underweight and above which was designed as normal weight) and 25 kg/m^2. Ideally, this manuscript should specify how people with a BMI of 20 kg/m^2 and 25 kg/m^2 were categorized, not just those with a BMI of above or below these values.

Authors’ response

We greatly appreciate this insightful comment. We have corrected this vital mistake and used BMI classification according to World Health Organization guidelines (WHO, 2010), was also considered: underweight (BMI<18.5 kg/m^2), normal weight (18.5 kg/m^2≤BMI≤24.9 kg/m^2) and overweight/obese (BMI≥25 kg/m^2). Thank you.

References:

World Health Organization: WHO. (2010, May 6). A healthy lifestyle - WHO recommendations. https://www.who.int/europe/news-room/fact-sheets/item/a-healthy-lifestyle---who-recommendations

• One question that arose while reading the study was why the multi-categorical outcome variable of internet addiction severity was used instead of the quantitative internet addiction score. Was this choice made purely for novelty, or were there other considerations? It is conceivable that since the internet addiction score variable can take on a broader range of different values, it may contain more information that the models can learn from than the internet addiction severity categorical variable, and hence it may have greater predictive power.

Authors’ response

We sincerely appreciate your insightful comments. The choice to use a multi-categorical outcome variable for internet addiction severity instead of a quantitative score was driven by several methodological and clinical considerations, as evidenced by the broader research context:

Studies (Guo, et al., 2020; Van Rooij and Prause, 2014) show distinct clinical implications at different severity levels, such as moderate/severe internet addiction, which helps identify critical thresholds for intervention and prioritize actionable diagnoses.

For consensus challenges, researchers often used the multi-categorical internet addiction to capture nuanced differences (Guo, et al., 2020). This approach addresses the lack of diagnostic consensus (Van Rooij and Prause, 2014), and mirrors diagnostic practices for substance use disorders (Musetti, et al., 2016)

For interpretative advantages, we used the multi-categorical IA. Categorical variables simplify analyses like logistic regression, which are used to identify predictors of problematic use (Yildirim, et al., 2024). Continuous scores, while precise, may obscure clinically meaningful patterns (Guo, et al., 2020).

Earlier studies proposed multiple diagnostic criteria that emphasize behavioral and psychosocial impacts rather than raw scores. Categorizing severity allows direct comparisons with these frameworks and enhances cross-study validity (Van Rooij and Prause, 2014; Musetti, et al., 2016).

Therefore, the multi-categorical IA approach was not merely novel but aimed to improve clinical utility, address measurement inconsistencies, and facilitate actionable insights for prevention and treatment.

References:

Van Rooij, A. and Prause, N., 2014. A critical review of “Internet addiction” criteria with suggestions for the future. Journal of behavioral addictions, 3(4), pp.203-213.

Guo, W., Tao, Y., Li, X., Lin, X., Meng, Y., Yang, X., Wang, H., Zhang, Y., Tang, W., Wang, Q. and Deng, W., 2020. Associations of internet addiction severity with psychopathology, serious mental illness, and suicidality: large-sample cross-sectional study. Journal of medical Internet research, 22(8), p.e17560.

Musetti, A., Cattivelli, R., Giacobbi, M., Zuglian, P., Ceccarini, M., Capelli, F., Pietrabissa, G. and Castelnuovo, G., 2016. Challenges in internet addiction disorder: is a diagnosis feasible or not?. Frontiers in psychology, 7, p.842.

Yildirim Demirdöğen, E., Akinci, M.A., Bozkurt, A., Bayraktutan, B., Turan, B., Aydoğdu, S., Ucuz, İ., Abanoz, E., Yitik Tonkaz, G., Çakir, A. and Ferahkaya, H., 2024. Social media addiction, escapism and coping strategies are associated with the problematic internet use of adolescents in Türkiye: a multi-center study. Frontiers in Psychiatry, 15, p.1355759.

• It is also important to note that the data does not support the first sentence of the Conclusion section. Namely, "The study results highlight a concerning rise in IA among university students in Bangladesh, posing significant risks to mental health, academic performance, and future professional careers." The use of a convenience snowball sample means that the data cannot reliably determine the prevalence of internet addiction among Bangladeshi university students or whether it is increasing.

Authors’ response

Thank you for raising a valid and important point regarding the limitations of the study's methodology, therefore we reframed the first sentence of the conclusion section to reflect the data's limitations, such as:

“The study results highlight a concerning level of internet addiction among university students in Bangladesh, which may pose risks to mental health, academic performance, and future professional careers. However, due to the use of a convenience snowball sample, these findings cannot be generalized to all Bangladeshi university students, and further research using more representative sampling methods is needed to assess the true prevalence and trends of internet addiction in this population.”

• The manuscript also does not discuss potential opportunities for further research arising from this work, which should ideally be included.

Authors’ response

Thank you for your valuable suggestion. We have revised the manuscript and included potential opportunities for further research (page # 25).

• Lastly, it would be helpful to include a table comparing the demographic breakdown of the sample to that of all Bangladeshi university students at the time of the study (or at least the universities sampled). This would allow readers to assess how representative the sample is. This is, of course, assuming that data on the demographic breakdown of all Bangladeshi university students is available (such as from Bangladesh's statistics bureau, or from the country's department of education)

Authors’ response

Thank you for your valuable comment. Unfortunately, Bangladesh’s statistics bureau and the education sector do not have well-developed records on the demographic breakdown of university students. Due to the unavailability of such data, we are unable to compare our sample with the overall Bangladeshi university student population. Additionally, as this study was conducted based on our research interest without external funding, we relied on primary data collection to the best extent possible.

Reviewer #2 comment:

Comments to the Author:

Introduction

• comment 1: The definition of Internet Addiction (IA) is comprehensive, but the sentence "This leads to minimizing offline time..." could be restructured for better readability. Consider breaking it into two sentences for clarity.

Example: "IA is defined as the inability to control the urge for excessive internet use. This often results in reduced offline activities and significant anxiety or aggression when internet access is restricted."

Authors’ response

Thank you for your valuable suggestion. We have revised the sentence for better readability and clarity (page #3).

• comment 2: The transition between the global context and Bangladesh-specific data could be smoother. Consider introducing the Bangladesh context with a linking sentence like, "This global phenomenon is also evident in Bangladesh, where..."

Authors’ response

Thank you for your suggestion. By incorporating a linking sentence, we have improved the transition between the global context and Bangladesh-specific data. The revised text now reads: 'This global phenomenon is also evident in Bangladesh, where the increasing number of internet users raises concerns about its impact on university students' (page #3).

• comment 3: The impact of IA on academic life is repeated in multiple places. You might consolidate these points for better focus and avoid redundancy.

Authors’ response

Thank you for your valuable feedback. We have reviewed the manuscript and consolidated the discussion on the impact of IA on academic life to avoid redundancy (page #3).

• comment 4: The comparison of internet usage time ("extra 0.64 hours per session...") could be more impactful with context or comparison to average usage.

Authors’ response

Thank you for your insightful suggestion. We have revised the comparison of internet usage time to provide better context by including a reference to average usage among university students (page#3).

comment 5: The connection between IA's effects and the Sustainable Development Goal (SDG) 3 is significant but could be made more prominent. Expand slightly on how addressing IA aligns with SDG 3 goals.

Authors’ response

Thank you for your valuable suggestion. We have expanded the discussion on how addressing Internet Addiction (IA) aligns with Sustainable Development Goal (SDG) 3 (page #4).

• comment 6: The introduction of ML techniques is clear, but the connection between their use and the problem at hand could be stronger. Add a brief explanation of why ML techniques are uniquely suited for this analysis.

Authors’ response

Thank you for your valuable suggestion. We have strengthened the connection between the use of Machine Learning (ML) techniques and the problem at hand by briefly explaining why ML is well-suited for this analysis (page #5).

• comment 7: Use consistent terminology, such as "university students" instead of alternating between "students" and "university students."

Authors’ response

Thank you for your suggestion. We have carefully reviewed the manuscript and revised it to ensure consistency in terminology by using 'university students' throughout, instead of alternating between 'students' and 'university students’.

• comment 8: Ensure all acronyms like ML, IA, DT, RF, SVM, and LR are clearly defined when first mentioned.

Authors’ response

Thank you for your suggestion. We have carefully reviewed the manuscript to ensure that all acronyms are clearly defined when first introduced.

Data Sources and Study Design

• Clarity: The use of "snowball sampling" and "convenient sampling" together might confuse readers. Clari

---

## [Decision Letter · Decision Letter 1]

An illustration of multi-class roc analysis for predicting internet addiction among university students

PONE-D-24-30985R1

Dear Dr. Rahman,

We’re pleased to inform you that your manuscript has been judged scientifically suitable for publication and will be formally accepted for publication once it meets all outstanding technical requirements.

Kind regards,

Jyotir Moy Chatterjee

Academic Editor

PLOS ONE

Additional Editor Comments (optional):

Reviewers' comments:

Reviewer's Responses to Questions

**Comments to the Author**

Reviewer #1: All comments have been addressed

Reviewer #2: All comments have been addressed

Reviewer #3: (No Response)

2. Is the manuscript technically sound, and do the data support the conclusions?

Reviewer #1: Yes

Reviewer #2: Yes

Reviewer #3: No

3. Has the statistical analysis been performed appropriately and rigorously?

Reviewer #1: Yes

Reviewer #2: Yes

Reviewer #3: I Don't Know

4. Have the authors made all data underlying the findings in their manuscript fully available?

Reviewer #1: Yes

Reviewer #2: Yes

Reviewer #3: No

5. Is the manuscript presented in an intelligible fashion and written in standard English?

Reviewer #1: Yes

Reviewer #2: Yes

Reviewer #3: No

Reviewer #1: I believe the authors have addressed my feedback very well. I have noticed that the titles of figures and tables do not always end in a full stop (or period). This is the case for Table 1, Figure 1, Table 2 and Figures 3-8.

Reviewer #2: Accepted. no need more revision. Thank you for submitting your manuscript. Based on my assessment, I have no specific concerns regarding research ethics, dual publication, or related publication ethics issues. The manuscript appears to adhere to standard ethical guidelines for academic research and publishing.

Reviewer #3: The paper does not adhere to the required formatting guidelines and appears to be neither properly structured nor compliant with the specified submission standards. I recommend reviewing the formatting instructions carefully and revising the document accordingly to ensure it meets the expected academic and presentation criteria.

**Do you want your identity to be public for this peer review?** For information about this choice, including consent withdrawal, please see our Privacy Policy

Reviewer #1: **Yes: ** Brenton Horne

Reviewer #2: No

Reviewer #3: No

---

## [Editor Report · Acceptance letter]

PONE-D-24-30985R1

PLOS ONE

Dear Dr. Rahman,

I'm pleased to inform you that your manuscript has been deemed suitable for publication in PLOS ONE. Congratulations! Your manuscript is now being handed over to our production team.

Kind regards,

on behalf of

Mr. Jyotir Moy Chatterjee

Academic Editor

PLOS ONE